# Discovery of Novel Trace Amine-Associated Receptor 5 (TAAR5) Antagonists Using a Deep Convolutional Neural Network

**DOI:** 10.3390/ijms23063127

**Published:** 2022-03-14

**Authors:** Carlotta Bon, Ting-Rong Chern, Elena Cichero, Terrence E. O’Brien, Stefano Gustincich, Raul R. Gainetdinov, Stefano Espinoza

**Affiliations:** 1Central RNA Laboratory, Istituto Italiano di Tecnologia (IIT), 16132 Genova, Italy; carlotta.bon@iit.it (C.B.); stefano.gustincich@iit.it (S.G.); 2Atomwise Inc., San Francisco, CA 94103, USA; ting-rong@atomwise.com (T.-R.C.); terry@atomwise.com (T.E.O.); 3Department of Pharmacy, University of Genoa, Viale Benedetto XV, 16132 Genoa, Italy; cichero@difar.unige.it; 4Institute of Translational Biomedicine, Saint-Petersburg State University, 199034 St. Petersburg, Russia; gainetdinov.raul@gmail.com; 5Saint-Petersburg University Hospital, Saint-Petersburg State University, 199034 St. Petersburg, Russia; 6Department of Health Sciences and Research, Center on Autoimmune and Allergic Diseases (CAAD), University of Piemonte Orientale (UPO), 28100 Novara, Italy

**Keywords:** trace amine-associated receptor 5, antagonist, AtomNet, BRET

## Abstract

Trace amine-associated receptor 5 (TAAR5) is a G protein-coupled receptor that belongs to the TAARs family (TAAR1-TAAR9). TAAR5 is expressed in the olfactory epithelium and is responsible for sensing 3-methylamine (TMA). However, recent studies showed that TAAR5 is also expressed in the limbic brain regions and is involved in the regulation of emotional behaviour and adult neurogenesis, suggesting that TAAR5 antagonism may represent a novel therapeutic strategy for anxiety and depression. We used the AtomNet^®^ model, the first deep learning neural network for structure-based drug discovery, to identify putative TAAR5 ligands and tested them in an in vitro BRET assay. We found two mTAAR5 antagonists with low to submicromolar activity that are able to inhibit the cAMP production induced by TMA. Moreover, these two compounds also inhibited the mTAAR5 downstream signalling, such as the phosphorylation of CREB and ERK. These two hits exhibit drug-like properties and could be used to further develop more potent TAAR5 ligands with putative anxiolytic and antidepressant activity.

## 1. Introduction

Trace amines are a class of endogenous chemicals present at low levels in the body, both in the periphery and the brain of vertebrates, including mammals [1]. Classic trace amines include beta-phenylethylamine (PEA), p-tyramine, tryptamine, and octopamine [2]. Originally believed to be inactive compounds and closely related by-products of endogenous monoamines such as dopamine and serotonin, the role of trace amines was re-evaluated when the trace amine-associated receptors (TAARs) family was discovered [3,4]. TAARs are a group of G protein-coupled receptors (GPCR) consisting of nine subfamilies with different genes and pseudogenes present in different species. For example, in humans, there are 6 genes (TAAR1, TAAR2, TAAR5, TAAR6, TAAR8 and TAAR9) and 3 pseudogenes (TAAR3, TAAR4 and TAAR7), while in mice, there are 15 functional genes [1]. Initial studies showed the expression of TAAR1 and other TAARs in some human and rodent brain regions. Except for TAAR1, all the other TAARs are expressed in the olfactory epithelium and form a new class of olfactory receptors that sense volatile amines involved in innate behaviours [5]. However, several lines of evidence strengthen the initial finding of a modest but widespread expression of different TAARs in the rodent and human brain. TAAR1 is the most studied member of the family and is activated not only by trace amines but also by amphetamines and other psychotropic compounds [3]. It modulates the dopamine system by influencing D2 dopamine receptor activity and the firing of dopaminergic neurons [6,7,8,9,10,11]. TAAR1 knockout (TAAR1-KO) mice display an enhanced behavioural and neurochemical effect induced by dopaminergic compounds [12,13,14]. Thus, TAAR1 has become a promising target for pharmacotherapy, and recent clinical trials suggest a potential use for schizophrenia of TAAR1 agonists with a new mechanism of action not involving blockade of D2 dopamine receptors [15].

TAAR5 is another receptor recently described as expressed in the brain at low levels, similarly to TAAR1. In mice, we showed that TAAR5 is present in limbic regions such as the amygdala, the entorhinal cortex, the nucleus accumbens, and the thalamic and hypothalamic nuclei [16], confirming previous reports that show TAAR1 and TAAR5 co-expression in the amygdala [17]. Transcriptomic data from human samples also confirm the presence of TAAR5 in limbic brain regions and highlight that TAAR5 is one of the most expressed TAARs in the central nervous system [18,19]. TAAR5 regulates emotional behaviour in mice, and TAAR5-KO mice show anxiolytic and antidepressant-like behaviours [16]. Furthermore, TAAR5 seems to modulate the serotonin system since TAAR5-KO mice have altered serotonin levels in different brain regions and enhanced functions of the 5-HT_1A_ receptor [16]. In addition, these mutants display somewhat elevated dopamine levels and increased adult neurogenesis in the subventricular and subgranular zones [20]. Recent data also show involvement in sensorimotor functions, and modulation of the electrocorticogram suggesting involvement of TAAR5 in cognitive processes such as attention and motivation [21]. We also have evidence on the role of TAAR5 in cognition, particularly in decision making and cognitive flexibility [22]. This evidence makes TAAR5 an attractive new target for drug discovery, especially for mood disorders and possibly for cognitive impairment.

The endogenous ligand, at least for the olfactory TAAR5, 3-methylamine (TMA) is a tertiary amine present in the rodent urine and spoiled food [5,23]. TMA has interesting functions in rodents, being attractive for mice and repulsive for rats [24]. In humans, the TMA is present in some spoiled food, and, for instance, it is found in degraded fish with an unpleasant odour. Interestingly, TAAR5 mutations in humans lead to impaired sensing of fish odour [23]. Just two other TAAR5 agonists have been identified. A-NETA, which is known as an inhibitor of acetyl-cholinesterase, shows good activity in activating TAAR5 and has interesting in vivo properties in rodents, such as producing a behaviour consistent with psychosis-related cognitive deficits [25,26]. 3-iodothyronamine (T_1_AM), an endogenous derivative of the thyroid hormone, activates TAAR1, but at the same time, it behaves as an agonist and inverse agonist toward mTAAR5 and hTAAR5, respectively [17]. A homology model for TAAR5 has been described and by screening a library of compounds targeting the serotonergic system it was possible to find the first two TAAR5 antagonists [27].

However, more selective and potent ligands are necessary to study the pharmacology of TAAR5 and to understand its role in brain physiology better. Given the potential application of TAAR5 antagonists in mood disorders, finding novel ligands with new chemical structures and better pharmacodynamics and pharmacokinetic profiles would be a step forward in psychopharmacology.

This study used AtomNet^®^ model, the first deep learning neural network for structure-based drug discovery, to identify putative TAAR5 ligands and tested them in an in vitro BRET assay. The methodological steps taken in this paper are depicted in Figure 1.

We found two antagonists, namely compounds **1** and **2** (chemical structure not shown, patent pending), with low to submicromolar activity that could antagonize the intracellular signalling mediated by TAAR5 activation. Furthermore, in silico prediction suggests a good pharmacokinetics (PK) profile of these compounds that may be considered good hits for further development in the future.

## 2. Results

### 2.1. Homology Modeling and In Silico Screening

In our previous work, we constructed the mTAAR5 homology model and screened it against an in-house focused library synthesized for 5HT_1A_ receptor and mTAAR1 screening [27]. We found two hits, **1a** and **2a** (see the Appendix A from [27]), that were both characterized as antagonists of mTAAR5 with uM activity. This study intends to explore novel chemical matters from a much larger and diverse chemical library using the AtomNet^®^ model, a structure-based deep convolutional neural network virtual screening technology developed by Atomwise. In the absence of published crystal structures, we built a homology model of mTAAR5 based on the recently published homologous protein Meleagris gallopavo β1-adrenoceptor (PDB ID: 6IBL, sequence identity of 35.59%) [28]. The constructed mTAAR5 model was superimposed to the template structure (Figure 2a), displaying a backbone RMSD value of 0.564 Å. No significant outliers were observed when inspecting the backbone conformation with the Ramachandran plot. (Appendix A). Using the constructed homology model, we performed a virtual screen against Enamine In-Stock HTS library with approximately 2 million commercially available, drug-like small molecules that yielded a chemically diverse set of 94 high-scoring predicted hits. As shown below, we successfully identified two mTAAR5 antagonists (compound **1** and **2**) with low micro-molar potency out of 94 tested compounds. Compounds **1** and **2** were ranked 247th and 287th out of the 2,273,009 compounds we screened, respectively.

### 2.2. Library In Vitro Screening

The selected compounds were initially tested using an in vitro approach. TAAR5 is a GPCR coupled to stimulatory G protein, and its activation evokes the production of cAMP. Therefore, in addition to rho-TAAR5, HEK293 cells were transfected with a cAMP BRET biosensor that changes the light emission accordingly to the cAMP fluctuation [29]. TMA was used as a reference rho-TAAR5 agonist at the concentration of 10 µM. The activity of 94 compounds was measured using the BRET-based assay and tested at 10 µM either for agonistic or antagonistic activity, followed by a dose–response assessment for those that have been found to be active.

No compounds displayed agonistic activity (data not shown). However, two molecules (compounds **1** and **2**) were characterized by a TAAR5 antagonist activity. Then, we performed a dose–response experiment, using the same compounds with a concentration range from 10 nM until 100 μM. The calculated IC50 values of the two antagonists were 2.8 ± 0.75 μM and 1.1 ± 0.92 μM, respectively (Figure 3).

### 2.3. TAAR5 Antagonists Inhibit ERK and CREB Phosphorylation

To further investigate TAAR5 downstream effectors, we used HEK293 cells expressing rhoTAAR5 and looked for signalling molecules known to be activated by an increase of cAMP, such as ERK and CREB. First, we performed a time course treating the cells with TMA at 10 μM and then lysing the cells at different time points. As shown in Figure 4, TMA activates ERK (a) and CREB (b) phosphorylation with the maximum effect at 5 and 15 min, respectively. After determining the optimal activation timing, we evaluated the antagonistic activity of compounds **1** and **2** by testing the efficacy in blocking ERK (c) and CREB (d) phosphorylation. As shown in Figure 4b, both molecules at 10 µM prevented the phosphorylation of both ERK and CREB.

### 2.4. In Silico Prediction of ADME Properties

During the last years, the drug discovery process relied on the in silico prediction of absorption, distribution, metabolism, excretion properties (ADME). Applying computational approaches to gain information on the pharmacokinetics (PK) and toxicity profile of ligands accelerated the lead optimization process [30,31]. Herein, we performed a computational prediction of PK properties shown by the newly screened compounds **1** and **2** compared to the previously developed mTAAR5 antagonists **1a** and **2a**. In order to highlight any putative violation of the Lipinski’ rule [32], we evaluated in silico the logarithmic ratio of the octanol–water partitioning coefficient (cLogP), the molecular weight (MW) of all the antagonists herein discussed, their number of H-bonding acceptor (HBA) and donor moieties (HBD), their number of rotatable bonds (nRot_bond) and the related topological polar surface area (TPSA) (Table 1).

While Lipinski’s rules suggest for compounds exhibiting MW < 500, cLogP < 5, HBA < 10 and HBD < 5, those by Veber are related to drug bioavailability thanks to nRot_bonds ≤10, the total number of H-bonding atoms (as the sum of HBD and HBA) < 12 and TPSA ≤ 140 Å^2^. Based on the in silico evaluation, all the screened compounds fulfil all the suggested Lipinski’s rule and Veber’s rule, as shown by the previously described **1a** and **2a**. Prediction of ADME properties was also performed, taking into account the human intestinal absorption (HIA), the ability to pass the blood–brain barrier (BBB) as predicted by LogBB (representative of Extent of brain penetration) and LogPS (representative of Rate of brain penetration). In addition, the volume of distribution (Vd), the role played by plasmatic protein binding (%PPB), and the ligand affinity toward human serum albumin (LogKa HSA) were all considered with the intent of determining the putative value of the oral bioavailability as a percentage (%F) (Table 2).

As shown in Table 2, all the newly developed mTAAR5 antagonists were predicted to be endowed with optimal absorption values (HIA = 100%) as well as the previous **1a** and **2a**. Notably, the novel **1** and **2** experienced more promising plasmatic protein binding values (%PPB = 71–77%) and affinity toward the human serum albumin (logKa HSA = 3.43–3.90) in tandem with adequate Vd values (Vd = 2–4.4 L/Kg), if compared to **1a** and **2a** (%PPB = 97–98%; logKa HSA = 4.24–4.50; Vd = 4.3–6.9 L/Kg). As a consequence, **1** and **2** featured higher bioavailability values (%F = 99.5–%). As regards their ability to pass the BBB, all of them were predicted to successfully access the CNS, as shown in Figure 5.

## 3. Discussion

TAAR5 is a member of the TAAR family known to be predominantly expressed in the olfactory system, and only recently several reports demonstrated its function in the CNS [16,21]. These first pieces of evidence suggest a potentially high impact of this receptor on brain physiology. However, TAAR5 pharmacology is poorly explored, and few ligands to study its activity exist. In our study, by using a novel AI-based algorithm, we discovered two TAAR5 antagonists with good potencies and a favourable PK profile.

Before cloning the TAARs family, TAAR5 was already known with the name of putative neurotransmitter receptor (PNR) [33]. Its expression in humans was reported at low levels in some brain areas, such as the amygdala, and in the periphery, such as the skeletal muscle. Thus, given its low level of expression, similarly to TAAR1 and other TAARs members, its study and even the clear demonstration of its presence in the brain were challenging. In fact, several groups that studied TAAR5 functions in the olfactory epithelium did not find it in the whole brain lysate, casting doubt on the presence of TAAR5 outside the olfactory system [5]. However, recently, our group demonstrated a low but distinct expression of TAAR5 in several brain regions, such as the amygdala, the entorhinal cortex, the nucleus accumbens, and the thalamic and hypothalamic nuclei, in accordance with previous reports showing TAAR1 and TAAR5 as being co-expressed in the amygdala [16,17]. Moreover, the analysis of transcriptomic data from two groups also demonstrated that TAAR5 is present in different regions of the human brain [18,19]. Importantly, TAAR5 seems to influence emotional behaviour. TAAR5-KO mice display an antidepressant-like phenotype, decreased anxiety [16], and increased adult neurogenesis [20]. Recent evidence also suggests an involvement of TAAR5 in cognition, mainly in attention, decision-making, and cognitive flexibility [22]. These data place TAAR5 as a novel potential pharmacological target for several psychiatric conditions with signs of depression, anxiety, and cognitive impairments.

Only a few TAAR5 agonists have been described. TMA is considered the endogenous TAAR5 ligand and is taken as a reference compound for in vitro screening of compound collections. TMA is present in mouse urine and activates TAAR5 in the olfactory epithelium with a pheromone-like property [24]. However, whether TMA is the endogenous ligands for TAAR5 expressed in other brain regions is still a matter of debate since it is virtually absent in these areas in normal conditions. Although with lower potencies, two other tertiary amines activate TAAR5, namely N-N-dimethylethylamine and N-methylpiperidine [5]. α-NETA, an inhibitor of acetyl-cholinesterase, potently activates TAAR5, and displays interesting in vivo properties in mice and rats, inducing psychotic-like states and influencing cortical parameters related to cognitive functions [25,26,34,35,36].

Based on evidence coming from TAAR5-KO mice, an antagonist may be an excellent ligand to test as a potential novel pharmacological tool. We already described a homology model of mTAAR5 and, by screening a serotonergic library, found the first two TAAR5 antagonists [27]. However, to identify more potent compounds with novel chemical structures, we built another homology model and used the AtomNet^®^ model to virtually screen against a more extensive, chemically diverse, drug-like library. Using this approach, we selected 96 compounds covering diverse chemical space and tested in vitro in an already validated BRET assay. As a result, we found two promising hits with low micromolar IC50 that could abolish TMA activity. The discovered hits are chemically distinct from the antagonists we discovered from our in-house library synthesized for GPCR projects. As seen before [37], this result demonstrated that the AtomNet^®^ model is able to discover novel scaffolds, even when on-target data are scarce or unprecedented. In addition, this work demonstrated the ability of AtomNet^®^ model on finding hits without available crystal structures or a high sequence identity template.

We compared the homology model built in this study with the AlphaFold predicted mTAAR5 structure (Appendix A). The AlphaFold predicted structure of mTAAR5 was obtained from AlphaFold Protein Structure Database [38,39] (AlphaFold DB version 30 June 2021, created with the AlphaFold Monomer v2.0 pipeline) using uniport ID: Q5QD14 as the query. The backbone RMSD value between the two structures is 1.88 Å. The residues composing the binding pocket from the transmembrane domain were aligned relatively well. However, we observed side-chain misalignments on ECL2 (extracellular loop 2) between the homology model and the AlphaFold structure (Appendix A). The ECL2 from the AlphaFold structure adopted an “inward” conformation resulting in a smaller binding site than the homology model structure (Appendix A). Specifically, F196 and L194 face upward toward the extracellular side in the homology model while facing inward in the AlphaFold structure (Appendix A). Placing the bond ligand found in the structure (PDB ID: 6IBL) onto the AlphaFold predicted structure resulted in a steric clash with L194 (Appendix A). Analyzing the pLDDT score provided by AlphaFold DB, the ECL2 region was categorized as “Low Confidence” and should be treated with caution (Appendix A). In fact, it is reported that some AlphaFold structures are in a non-ligand bound state with empty pocket and are, therefore, not ideal for virtual screening [40]. In addition, it has been shown that, in some cases, the protein–ligand interaction region was not modeled as high confidence by AlphaFold [41]. Therefore, checking the pLDDT value around the binding site can be a good practice to evaluate the feasibility of using AlphFold structure for virtual screening.

We then studied the intracellular signalling mediated by TAAR5. GPCRs can signal through different proteins and kinases [42], and cAMP production stimulated by Gαs often lead to CREB and ERK phosphorylation. As expected, TMA induced both ERK and CREB phosphorylation with different temporal dynamics. Interestingly, both antagonists prevent these effects, confirming their functionality in reducing cAMP levels. Whether TAAR5 can induce the phosphorylation of ERK and CREB through the cAMP/PKA pathway or also through the β-arrestin pathway remains to be determined. TAAR1 also induces similar signalling pathways and does not recruit strongly β-arrestin in vitro [43], although in vivo TAAR1-KO mice have an evident alteration of the AKT/GSK3 signalling pathway mediated by β-arrestin2, likely due to TAAR1 influence on D2 dopamine receptor [6,9].

Furthermore, we evaluated in silico their ADME properties to make a preliminary profile of their drug-likeness. Both compounds seem to have favourable PK, with a lower binding of plasma albumin compared to the previously discovered antagonist and a predicted good oral availability. Most importantly, the prediction of the BBB permeability is promising. Undoubtedly, the in vivo evaluation of the real PK and these compounds’ efficacies in modulating emotional behaviour is required. Nevertheless, our results may be a step forward in advancing the knowledge of TAAR5 pharmacology and TAAR5 ligand discovery. As for many psychiatric disorders, there is a strong need to discover and validate new targets for the pharmacotherapy of depression and anxiety. Indeed, patients with these disorders have been using essentially the same drugs in the last decades. Only the discovery of ketamine and its derivatives was a real breakthrough for this field recently. As TAAR1 might become a new target for schizophrenia since the discovery of D2 dopamine receptor blocking the action of antipsychotics more than 50 years ago, TAAR5 may emerge as a novel target for mood disorders.

## 4. Materials and Methods

### 4.1. Homology Modeling

At the onset of the virtual high-throughput screen for mTAAR5 modulators, no experimentally determined structures were publicly available. A search for template structures using SWISS-MODEL [44] found homologous structures in the sequence identity ranging from 29.73% (PDB ID: 7E32) to 39.22% (PDB ID: 7BZ2). Protein structures with sequence identity > 32% and X-ray crystal structure resolutions of <3 Å were chosen to build homology models using SWISS-MODEL. The homology model analysis includes factors such as alignment coverage, presence of bound ligand, resolution, backbone RMSD to the template structure, and the GMQE (Global Model Quality Estimate) value calculated from SWISS-MODEL. For the final assessment, we analyzed the Ramachandran plot using MolSoft ICM-Pro 3.8 software [45] for each generated homology model. Ultimately, we chose the homology model of mTAAR5 built on the homologous protein Meleagris gallopavo β1-adrenoceptor (PDB ID: 6IBL) with a sequence identity of 35.59%, resolution of 2.7 Å, alignment coverage of 88%, and GMQE value of 0.59 using SWISS-MODEL.

### 4.2. Virtual Screening Method

Virtual high-throughput screening was performed using the AtomNet^®^ model for structure-based drug design trained to predict protein–ligand binding [46,47]. A single global AtomNet^®^ model was used to predict binding affinity for small molecules with protein targets. The model was trained with K_i_, K_d_ and IC50 values of several millions of small molecules and thousands of protein structures from a variety of protein family. The experimental data and the structural information were curated from public database and proprietary sources. Next, the model was applied prospectively for novel binding sites with unprecedented ligands. Because the AtomNet^®^ model is a single global model, it helped prevent model overfitting. The model training proceeded as the following three-step procedure. First, the binding site of a given target was defined using a flooding algorithm [48] based on an initial seed. The initial starting point of the flooding algorithm might be determined by either a bound ligand annotated in the PDB database, or crucial residues as revealed by mutagenesis studies, or identification of catalytic motifs previously reported. Second, we shifted the coordinates of the protein–ligand complex to a three-dimensional Cartesian system with an origin at the center-of-mass of the binding site. To prevent the neural network from memorizing a preferred orientation of the protein structure, we performed data augmentation by randomly rotating and translating the protein structure around the center-of-mass of the binding site. Third, we sampled multiple poses within the binding site for each ligand. Each pose represents a putative protein–ligand complex; therefore, our method does not require experimental co-complexes for either training or prediction. Each generated co-complex was then rasterized into a fixed-size regular three-dimensional grid, where the values at each grid point represent the structural features that are present at each grid point. Similar to a photo pixel containing three separate channels representing the presence of red, green, and blue colors, the grid points represented the presence of different atom types. These grids served as the input to a convolutional neural network and defined the receptive field of the network. We used a network architecture of a 30 × 30 × 30 grid with a 1 Å spacing for the input layer, followed by five convolutional layers of 32 × 3^3^, 64 × 3^3^, 64 × 3^3^, 64 × 3^3^, 64 × 2^3^ (number of filters × filter-dimension), and a fully connected layer with 256 ReLU hidden units. The scores for each pose in the ensemble were then combined through a weighted Boltzmann averaging to produce a final score. These scores were compared against the experimentally measured pK_d_, pK_i_ or pIC50 of the protein and ligand pair, and the weights of the neural network were adjusted to reduce the error between the predicted and experimentally measured affinity using a mean-square-error loss function. Training was performed using the ADAM [49] adaptive learning method, the backpropagation algorithm, and mini-batches with 64 examples per gradient step.

The screening site was centered around the mTAAR5 residues S91, R94, T111, D114, T115, C118, L119, Y165, C192, Q193, L194, F196, W200, G201, L203, N204, A207, W265, F268, T272, F287, D288, I291, Y295 (determined as the potential binding site based on alignment to the template complex (PDB ID: 6IBL)) (See Figure 2b–d). The Enamine In-Stock HTS library of approximately 2 million small molecule compounds were prepared and screened, as described previously [46] using an ensemble of protein–ligand conformations. Each of the 2 million molecules were scored and ranked by the AtomNet^®^ model. The top-scoring 30,000 compounds were filtered for MW of >200 Da and then filtered with proprietary SMARTS patterns to eliminate undesired chemical moieties. SMARTS patterns for similar applications are available in the public domain, such as [50]. The remaining compounds were further clustered by ECFP4 fingerprint with Tanimoto coefficient similarity cutoff of 0.35 using the Butina clustering algorithm [51]. The final set of 2556 cluster representatives were further filtered using the following criteria: molecular weight of <500 Da, log P < 5, PAINS score < 0.6, toxicology score of <2.0, “bad groups” = none using MolSoft ICM-Pro 3.8 software. Of the remaining compounds, 94 compounds that passed an internal quality control procedure were purchased for experimental testing.

### 4.3. Cell Culture and Transfection

Human embryonic kidney 293 cells (HEK293) were maintained in high-glucose GlutaMAX™ Dulbecco’s modified Eagle’s medium (Gibco™) supplemented with 10% (*vol*/*vol*) of FBS and 1% penicillin/streptomycin at 37 °C in a humidified atmosphere at 95% air and 5% CO_2_. For the bioluminescence resonance energy transfer (BRET) experiments, cells were plated in 10 cm dishes 24 h prior the transient transfection of 7 μg of rho-TAAR5 and 7 μg of EPAC using lipofectamine 2000 (Waltham, MA, USA). Five hours after transfection, cells were plated in poly-D-lysine-coated 96-well microplates (well-assay white plate with clear bottom, North Carolina USA) at a density of 70,000 cells per well in OPTI-mem I (Gibco™) and then cultured for an additional 24 h. For the phospho-ERK and phospho-CREB assays, cells were plated in 6-well plates (NY, US) 24 h prior the transient transfection of 1 µg of rho-TAAR5 using lipofectamine 2000 (Invitrogen) and then cultured for an additional 24 h.

### 4.4. BRET Assay

BRET experiments were performed as described previously [29]. All the compounds were tested at the initial concentration of 10 μM. For the evaluation of the agonistic activity, the plate was read immediately after the addition of the compounds for approximately 20 min. For the evaluation of the antagonistic activity, the compounds were added 5 min before the addiction of the control TAAR5 agonist, TMA, and read for approximately 20 min. For the ones that were active, a dose–response assessment was performed using different concentration of the antagonist. The IC_50_ was then calculated by measuring the effect of the compounds against the effect of TMA at 10 μM. Readings were collected using a Tecan Infinite instrument that allows the sequential signals integration detected in the 465 to 505 nm and 515 to 555 nm windows. EPAC BRET biosensor was used to monitor cAMP levels. Increased cAMP specifically reflects the decrease in the BRET ratio. The acceptor/donor ratio was calculated as previously described [52]. The curve was fitted using non-linear regression and one site-specific binding with GraphPad Prism 9 software. The data are representative of at least 3 independent experiments and are expressed as means ± SEM.

### 4.5. Phospho-ERK and Phospho-CREB Assays

HEK293 cells expressing rho-TAAR5 were treated, 24 h post-transfection, with TMA at 10 μM and lysed at different time points (time course experiment) to determine the optimal pERK and pCREB activation time. Then, the antagonistic activity of active compounds were evaluated at 5 (pERK) and 15 (pCREB) minutes. Vehicle and TMA treatment were used as negative and positive control respectively.

### 4.6. Western Blot Analysis and Antibodies

Cells were lysed with RIPA buffer supplemented with protease (Roche) and phosphatase (Waltham, MA, USA) inhibitors. After 10 min of incubation on ice, lysates were centrifugated for 10 min at 13,000 rpm and supernatant was collected for total protein concentration assay (BCA protein assay, Thermo Scientific). A measure of 25 μg of protein extract was separated on 10% SDS/PAGE and transferred on nitrocellulose membrane. The anti-phospho-ERK1/2 (Thr-202/Tyr204) and anti-phospho-CREB (Ser-133) were purchased from Cell Signalling Technology. All the primary antibodies were incubated overnight, rocking at 4 °C. Appropriated secondary antibodies and chemiluminescent reagents were used. For quantitative analysis, housekeeping proteins (actin) were used as loading controls for phosphoprotein signals. The obtained results were normalized to the respective vehicle control.

### 4.7. In Silico ADME Properties Prediction

The prediction of all the parameters herein reported as related to ADME properties was performed by means of the Advanced Chemistry Development (ACD) Percepta platform (ACD/Percepta Platform, Advanced Chemistry Development, Inc., Toronto, ON, Canada, 2015 (v14.0.0) [53], thanks to the training libraries implemented in the software. These refer to different series of compounds whose pharmacokinetic properties have been experimentally evaluated.

### 4.8. Statistical Analysis

The data are presented as mean ± SEM and analyzed using two-tailed Student *t*-test or one-way ANOVA followed by Dunnet’s multiple comparison unless otherwise indicated. * *p* ≤ 0.05; ** *p* ≤ 0.01; *** *p* ≤ 0.001.

## Figures and Tables

**Figure 1 ijms-23-03127-f001:**
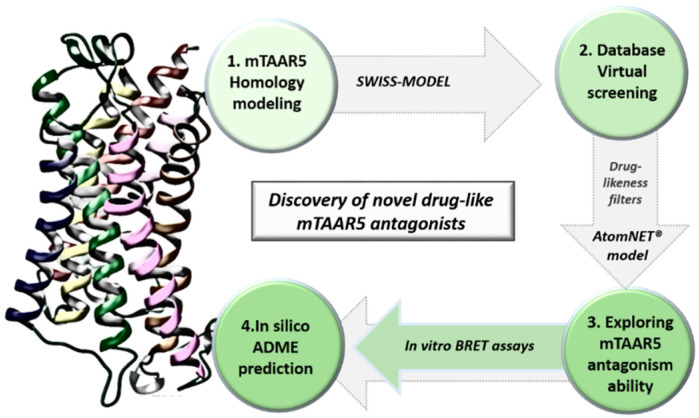
Scheme of the applied screening protocol followed by biological assays and in silico ADME predictive tools.

**Figure 2 ijms-23-03127-f002:**
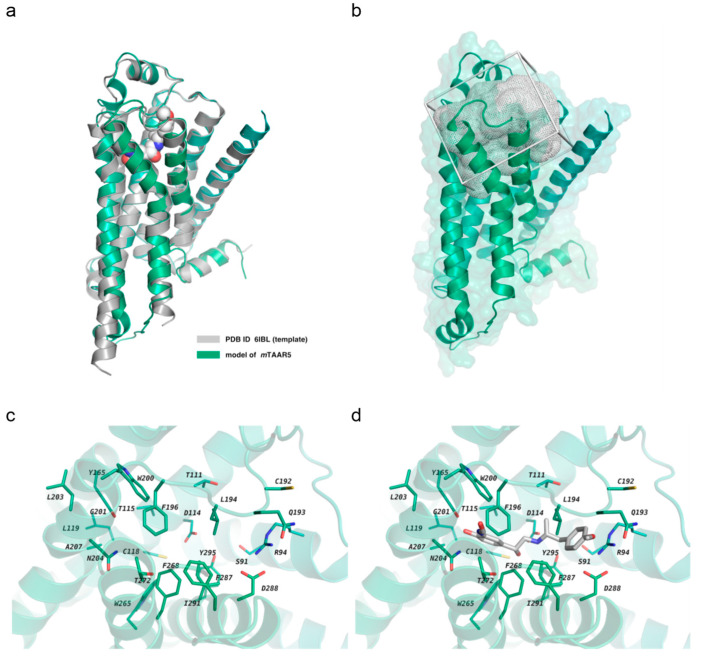
The homology model and the virtual screening sites of mTAAR5: (**a**) An overlay of the cartoon representation of constructed mTAAR5 model and homologous protein Meleagris gallopavo β1-adrenoceptor with bound ligand shown as a sphere (PDB ID: 6IBL). (**b**) The grid box covered the screening site shown as mesh. (**c**) A close-up view of relevant residues of the binding site from the intracellular side, shown as sticks and labeled; (**d**) the same site as in (**c**), but with the bound ligand found in the template structure (PDB ID: 6IBL) shown white.

**Figure 3 ijms-23-03127-f003:**
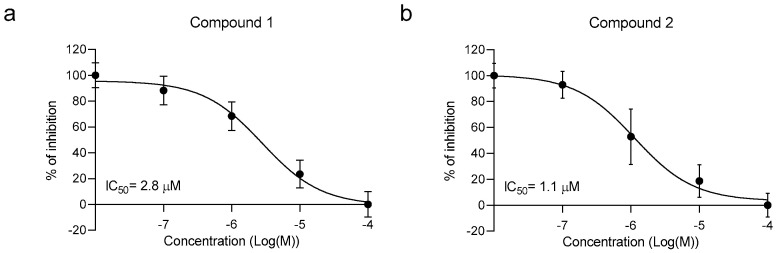
cAMP variation in cells co-expressing rho-TAAR5 and BRET EPAC biosensor. Cells were treated with the compounds at different concentrations and plotted as dose–response experiments. Non-linear regression with one site-specific binding is used to draw the curve using GraphPad Prism9. The data are plotted as a percentage of inhibition ± SEM of 3 independent experiments for compounds **1** (**a**) and **2** (**b**).

**Figure 4 ijms-23-03127-f004:**
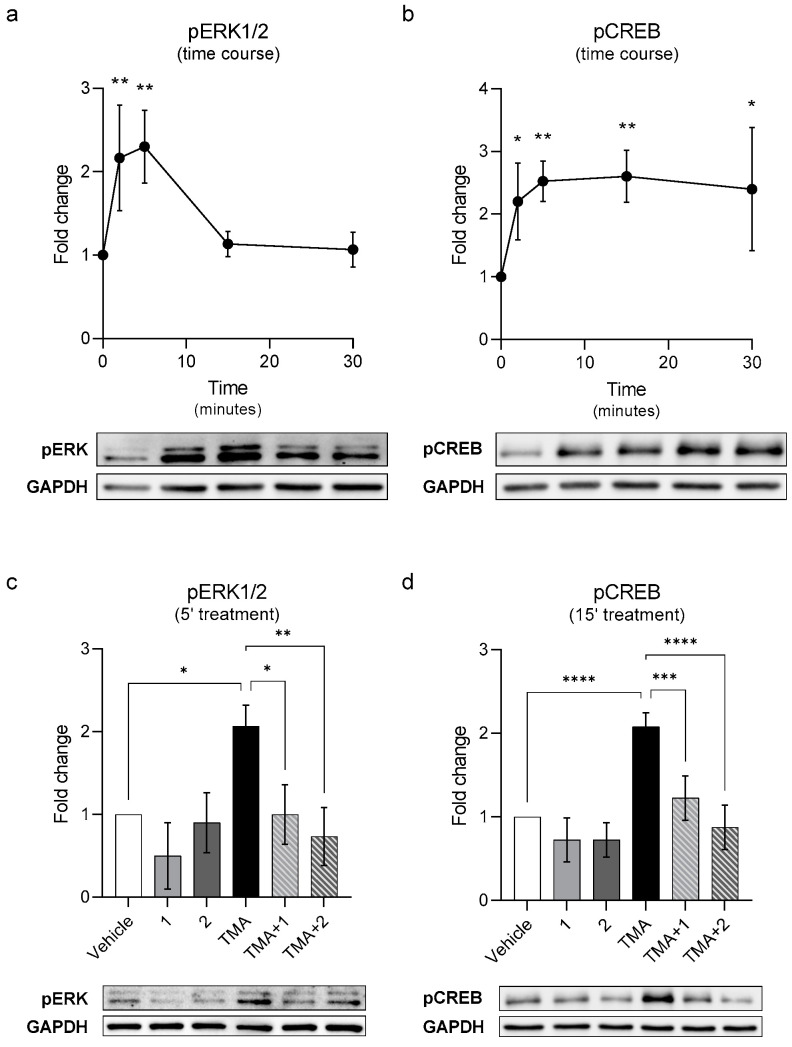
Evaluation of the antagonistic activity on ERK and CREB phosphorylation. HEK293 cells transfected with rho-TAAR5 were treated with TMA at 10 μM and lysed at different time points to assess ERK (**a**) and CREB (**b**) phosphorylation. After evaluating the best time point at which TMA induced an increase in pERK and pCREB, cells were treated with active compounds at 10 μM individually or in a set with TMA (**c**,**d**). Data represent means ± SEM. *n* = 3, 4 per group. * *p* < 0.05; ** *p* < 0.01; *** *p* < 0.001; **** *p* < 0.0001. One-way ANOVA followed by Dunnet’s multiple comparison test.

**Figure 5 ijms-23-03127-f005:**
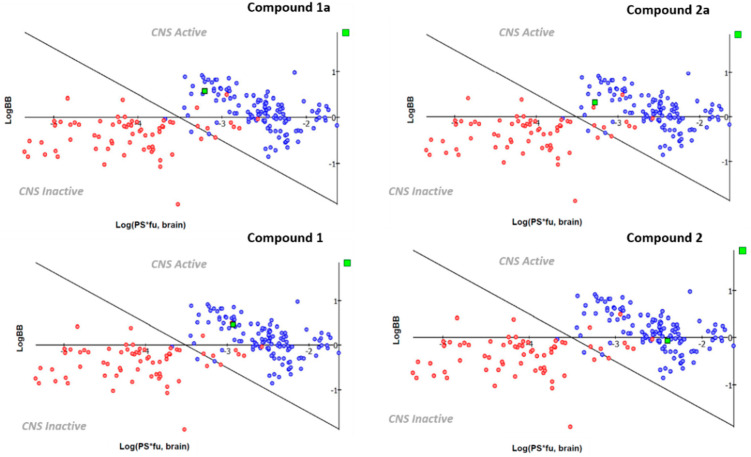
Scheme of the predicted BBB permeability featured by the previous **1a** and **2a** and by the novel **1** and **2** mTAAR5 antagonists. CNS inactive and active drugs, as reported in the ACD/Percepta training libraries, are reported in red and blue, respectively.

**Table 1 ijms-23-03127-t001:** Calculated descriptors related to the Lipinski’s rules and the Veber’s rules, referred to the **1a**, **2a** previously discovered mTAAR5 antagonists and to the newly screened compounds **1** and **2**. ^a^ cLogP as the logarithmic ratio of the octanol–water partitioning coefficient, ^b^ MW as the molecular weight of compounds, ^c^ HBA represents the number of H-bonding acceptor groups, ^d^ HBD represents the number of H-bonding donor groups, ^e^ nRot_bonds as number of rotable bonds, ^f^ TPSA represents the topological polar surface area.

Compound	cLogP ^a^	MW ^b^	HBA ^c^	HBD ^d^	nRot_bond ^e^	TPSA ^f^
**1a**	5.05	430	5	0	6	34.17
**2a**	4.67	405	5	1	9	48.95
**1**	3.07	350	4	1	7	35.58
**2**	1.64	343	6	1	5	54.04

**Table 2 ijms-23-03127-t002:** Calculated ADME parameters related to absorption and distribution properties as referred to 1a, 2a previously discovered mTAAR5 antagonists and to the newly screened 1 and 2. ^a^ HIA represents the human intestinal absorption, expressed as percentage of the molecule able to pass through the intestinal membrane. ^b^ Extent of brain penetration based on ratio of total drug concentrations in tissue and plasma at steady-state conditions. ^c^ Rate of brain penetration. PS represents permeability–surface area product and is derived from the kinetic equation of capillary transport ^d^ prediction of volume of distribution (Vd) of the compound in the body; ^e^ eplasmatic protein binding event; ^f^ ligand affinity toward human serum albumin; ^g^ oral bioavailability as a percentage.

Compound	HIA (%) ^a^	LogBB ^b^	LogPS ^c^	Vd (l/kg) ^d^	%PPB ^e^	LogKa HSA^f^	%F (Oral) ^g^
**1a**	100	0.56	−1.1	6.9	97%	4.50	89.3%
**2a**	100	0.31	−1.3	4.3	98%	4.24	99.3%
**1**	100	0.46	−1.8	4.4	77	3.43	99.5
**2**	100	−0.07	−1.9	2	71	3.90	99.5

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
