# Peer review of "Discovery of Novel Trace Amine-Associated Receptor 5 (TAAR5) Antagonists Using a Deep Convolutional Neural Network"

_ijms, 2022, doi:10.3390/ijms23063127_

Round 1

Reviewer 1 Report

Dear authors,

thank you for the manuscript. It can be considered interesting for the Journal's readership. However I have some major points.

  1. Please, provide detailed description of the AtomNet CNN and features that you used.
  2. Could you please describe all possible pharmacological and side effects for  TAAR5 antagonists? Are there any other efforts to find TAAR5 antagonists? it would be great to provide this information in the text.
  3. What is the rationale under selection of ERK and CREB as the molecules, which were influenced by TAAR5 antagonists? 

Reviewer 2 Report

Please refer to the attached document
